# Achieving Order in Disorder: Stabilizing Red Light-Emitting α-Phase Formamidinium Lead Iodide

**DOI:** 10.3390/nano13233049

**Published:** 2023-11-29

**Authors:** Aditya Narayan Singh, Atanu Jana, Manickam Selvaraj, Mohammed A. Assiri, Sua Yun, Kyung-Wan Nam

**Affiliations:** 1Department of Energy and Materials Engineering, Dongguk University—Seoul, Seoul 04620, Republic of Korea; aditya@dongguk.edu; 2Division of Physics and Semiconductor Science, Dongguk University, Seoul 04620, Republic of Korea; atanujanaic@gmail.com; 3Department of Chemistry, Faculty of Science, King Khalid University, Abha 61413, Saudi Arabia; mselvaraj@kku.edu.sa (M.S.); maassiri@kku.edu.sa (M.A.A.); 4Department of Advanced Battery Convergence Engineering, Dongguk University—Seoul, Seoul 04620, Republic of Korea; suanet50@gmail.com; 5Center for Next Generation Energy and Electronic Materials, Dongguk University—Seoul, Seoul 04620, Republic of Korea

**Keywords:** halide perovskite, stability, single particle imaging, phase transitions, water-stable

## Abstract

While formamidinium lead iodide (FAPbI_3_) halide perovskite (HP) exhibits improved thermal stability and a wide band gap, its practical applicability is chained due to its room temperature phase transition from pure black (α-phase) to a non-perovskite yellow (δ-phase) when exposed to humidity. This phase transition is due to the fragile ionic bonding between the cationic and anionic parts of HPs during their formation. Herein, we report the synthesis of water-stable, red-light-emitting α-phase FAPbI_3_ nanocrystals (NCs) using five different amines to overcome these intrinsic phase instabilities. The structural, morphological, and electronic characterization were obtained using X-ray diffraction (XRD), field emission scanning electron microscope (FESEM), and X-ray photoelectron spectroscopy (XPS), respectively. The photoluminescence (PL) emission and single-particle imaging bear the signature of dual emission in several amines, indicating a self-trapped excited state. Our simple strategy to stabilize the α-phase using various amine interfacial interactions could provide a better understanding and pave the way for a novel approach for the stabilization of perovskites for prolonged durations and their multifunctional applications.

## 1. Introduction

Few things are as widely agreed upon in the scientific community as the fact that the Earth’s atmosphere will continue to heat up if anthropogenic emissions of greenhouse gases, particularly carbon dioxide (CO_2_), are not significantly reduced towards zero [1]. To reduce CO_2_ emissions, achieve carbon neutrality, and prevent the global temperature from increasing by 2 °C this century, it is mandatory to achieve a 43% reduction in global emissions by 2030. Replacing fossil fuels with renewable energy (RE) technology is a major strategy to combat global warming. RE installations, particularly solar and wind, have steadily grown to two-thirds of global investments in electricity production [2]. In this race, photovoltaics (PVs) as an energy conversion technology has been the most cost-effective and indeed competitive with conventional energy technologies, even in moderate climates [3]. RE technologies are greatly appreciated due to their lack of CO_2_ emissions while converting solar radiation or airflow to usable forms of energy such as electricity. Thus, RE technologies are potent enough to reduce CO_2_ emissions by avoiding fossil fuel combustion during their operation.

Of all the RE technologies, including batteries [4,5], fuel cells [6,7], water splitting [8], etc., organic-inorganic halide perovskites (OIHPs) are prominent light harvesting materials due to their remarkable optical and photovoltaic properties [9,10,11,12]. Their tunable bandgaps, carrier diffusion length, carrier mobility, and other intrinsic properties have sparked a fury of research interest, intending to better understand their fundamental structure, motional behavior, and, most importantly, their stability [13].

Among OIHPs, formamidinium lead iodide (FAPbI_3_), an interesting material of the perovskite family (with a general representation of ABX_3_, A = organic/inorganic cation, B = metal cation, X = halide ion), has garnered intensive research interest due to its low band gap (1.48 eV) [14] and, most importantly, its larger organic cation (FA) relative to methyl ammonium (MA), thereby rendering a more symmetrical crystal structure, better light absorption near the infrared region, and elevated decomposition temperature that significantly enhances its thermal stability [15]. However, the phase purity of FAPbI_3_ is a major setback during its synthesis, hindering its wider applicability [16].

During its synthesis, FAPbI_3_ exhibits different phases with variations in temperature. Specifically, at room temperature (<300 K), the crystal structure adopts a trigonal non-perovskite δ-phase [17,18,19]. This trigonal non-perovskite yellow δ-phase is a one-dimensional (1D) needle-like structure, possessing a large band gap (2.43 eV), leading to a diminutive photoactivity [20,21]. While at a higher temperature (>330 K), it forms a black α-phase, which is luminescent, has a narrow band gap of 1.48 eV, is 3D, and holds several practical implications [22,23]. Because of its narrow band gap and enhanced thermal stability, α-phase FAPbI_3_ is considered a potential perovskite for high-performance single-junction perovskite solar cells [16]. However, this α-phase is only stable above 185 °C and readily transforms to the δ-phase at room temperature, particularly under humid conditions. Thus, it becomes imperative to stabilize α-FAPbI_3_ at room temperature to extract maximum benefits and push its wider applicability. The room-temperature instability of α-phase is linked to the rotational disorder of the FA cation and the huge anisotropic lattice strain, leading to high formation energies of α-phase FAPbI_3_ [24]. Thus, there is an urgent need to stabilize the α-phase by utilizing a well-designed synthesis protocol with minimum strain to gain the benefits of this phase in perovskite research.

To counter the above phase issues, several synthesis protocols have been used to obtain stable and pure α-phase, mainly by the wet-chemical method [25,26,27]. Before moving on to discuss the phase issues, it is essential to understand the crystal structure of this material. α-FAPbI_3_ is a 3D framework where octahedrons are connected by I^−^ anions and FA^+^ cations reside in 12-fold coordination (Figure 1). As the resultant distance between Pb-I (0.317 nm) is much shorter than the expected optimal distance (0.34 nm) given the radii of Pb^2+^ and I^−^ to be 0.12 and 0.22 nm, respectively, the α- phase is strained [14]. These strained α-phases spontaneously convert to non-perovskite δ-phases, destroying the benefits that could be extracted from them. Indeed, density functional theory reveals that the δ-phase is inherently more stable than the α-phase by 0.245 eV/f.u. [14]. The strained (111) plane becomes the driving factor for phase transition to δ-phase, which eventually increases in the presence of moisture, probably because the unstable organic cation hydrophilically interacts with water, releasing energy for stabilization [28]. Thus, it is a formidable task to stabilize the α-phase in humid conditions.

Stabilizing perovskites by surface functionalization is an established method that depends on anchoring large-sized organic molecules onto the perovskite’s surface. A metastable lead iodide perovskite was stabilized via surface functionalization by long-chain alkyl or aromatic ammonium cations anchored to the FAPbI_3_ structure [29]. Structural stability at room temperature is achieved by lowering the formation energy. In addition to quasi-2D and layered perovskite structures [30,31], 3D perovskites are widely explored due to their higher solar cell efficiencies than their 2D counterparts. However, 3D perovskites suffer from poor stability; hence, rigorous effort must be taken to stabilize them. Recently, the entropic stabilization method has been devised to stabilize newly developed (A)_1−x_(en)_x_(Pb)_1−0.7x_(X)_3−0.4x_ (where A = MA, FA; X = Br, I; MA = methylammonium; FA = formamidinium; en = ethylenediammonium), also referred to as “hollow” perovskites, due to excessive vacancies of Pb and X created due to the incorporation of en cations in the 3D network [32]. Stabilization using graphene nanosheets (NSs) functionalized with different chemical moieties has been an effective tool for perovskite nanocrystals [33,34]. While these methods have been effective in some cases, the presence of oxygen-containing functionalities can alter the growth process and result in perovskite materials with oxygen defects. Similarly, surface functionalization strategies such as phenylamine, iodine, and nitrogen-related strategies can significantly affect grain growth, making them unsuitable for certain applications [35,36,37,38]. Though other equally effective stabilization techniques use inorganic oxides such as SiO_2_, AlOx, and so on, they render the physical isolation of perovskites harder, making perovskites’ solution processability harder [39].

Herein, we report the synthesis of water-stable α-FAPbI_3_ following a top-down approach by encapsulating the stable bulk δ-FAPbI_3_ with various amines (aliphatic, aromatic, and so on). We then systematically study their crystal structure with X-ray diffraction (XRD) studies. To reveal the crystal morphologies and the electronic nature of the elements, we deploy field emission scanning electron microscopy (FESEM) and X-ray photoelectron spectroscopy (XPS), respectively. Photoluminescence (PL) spectroscopy and single-particle imaging were carried out to gain more insights into the optical properties. Thermogravimetric analysis (TGA) was also conducted to successfully demonstrate the thermal decomposition behavior. The structural stability of our perovskite, even in humid conditions, is ascribed to the hydrophobic nature of long chains of amines adhering to FAPbI_3_ by possibly converting to their respective iodide salts and integrating into the crystal structure of the perovskites.

## 2. Materials and Methods

### 2.1. Materials and Reagents

Basic lead (II) carbonate [(PbCO_3_)_2_.Pb(OH)_2_, 325 mesh], formamidinium acetate (HN=CHNH_2_.CH_3_COOH, 99%), hydroiodic acid (HI, 57 wt.% in H_2_O, distilled, stabilized, 99.95%), ethyl acetate (CH_3_COOCH_2_CH_3_, anhydrous, 99.8%), and a series of amines (oleylamine, ethylenediamine, N-methylaniline, 2-aminoethanol, and 4-4′-methylenebis) were purchased from Sigma-Aldrich (St. Louis, MO, USA).

### 2.2. Synthesis of δ-FAPbI_3_

A molar mass of basic lead carbonate (say 0.5 mmol, 0.388 gm) and formamidinium acetate (1.5 mmol, 0.156 gm) were added in HI (2 mL), and an immediately yellow precipitate appeared. To achieve full conversion, the precipitates were sonicated for 5 min and then washed with ethyl acetate (15 mL). The yellow precipitate was dried in an oven at 70 °C.

### 2.3. Synthesis of α-FAPbI_3_

In the typical synthesis of α-FAPbI_3_ capped with oleylamine, ethylenediamine, N-methylaniline, 2-aminoethanol, and 4-4′-methylenebis, we used varying amounts of different amines but with the same amount of HI. The amounts of basic lead carbonate and formamidinium were also kept identical. The molar amounts and other details used in the synthesis protocol are detailed in Table 1.

Briefly, the reaction mechanism for the synthesis of δ-FAPbI_3_ and then α-FAPbI_3_ could be understood as follows: firstly, the formamidinium acetate reacts with HI and forms FAI; subsequently, PbI_2_ and FAI react, forming a yellow-colored δ-FAPbI_3_. While for α-FAPbI_3_, we selected several amines and then dissolved them in the HI to form their respective iodide solutions. The final solution was then added directly to δ-FAPbI_3_ to form water-stable α-FAPbI_3_.

### 2.4. Characterization Methods

The crystal structure of the synthesized material was revealed by high-power powder X-ray diffraction (HP-PXRD) measurements using a Rigaku X-ray diffractometer with a 3-phase, 380 V, and 18 kW and equipped with Cu Kα radiation (λ = 1.54 Å). The samples for XRD analysis were prepared using the conventional technique of applying a thick film of powder samples onto a glass substrate. The XRD data was collected in the 2θ range, spanning from 5 to 50 degrees, with a scan rate of 2°/min.

X-ray photoelectron spectroscopy (XPS) measurements were conducted using a K-alpha instrument (from Thermo Fisher, Brighton, UK). Subsequently, the XPS data were processed and analyzed with the widely recognized XPS peak fitting software, CasaXPS version 2.3.22.

Particle morphologies were captured by SU8220 Cold FE-SEM, Hitachi High-Technologies, under an acceleration voltage of 10 kV, following the standard procedure for SEM sample preparation.

Photoluminescence spectra were taken with a Cary Eclipse fluorometer (Varian, Las Vegas, NV, USA) in solid-state.

Single-particle imaging. Photoluminescence single-particle imaging was conducted by Carl Zeiss utilizing the LSM 780 NLO microscope. The powder samples were evenly distributed on glass slides. Sample focusing was achieved through mechanical adjustments, employing both 10× air and 100× oil objective lenses. The experimental laser operated at 405 nm, and detection spanned the range of 410–700 nm (with a spectral resolution of 8.9 nm). For measurements, a GaAsP PMT detector comprising 32 channels was employed.

Thermogravimetric analysis (TGA) was performed using the Q500 model, TA. The heating rate was maintained at 10° per minute.

## 3. Results and Discussion

In this work, we first synthesize bulk δ-FAPbI_3_ and then α-FAPbI_3,_ capped with various amines, using water as a solvent, as detailed in the experimental section. A few previous studies used FA oleate, lead oleate, along with PbI_2_, capping ligand as oleylammonium iodide, and various other organic solvents to stabilize this perovskite [40]. However, we used economical precursors in our synthesis approach, which is helpful if bulk synthesis is required. For the synthesis of water-stable α-FAPbI_3,_ we selected several amines first dissolved in HI (detailed in the experimental section), and the obtained solution was then directly added to δ-FAPbI_3._ This addition brought a dark reddish precipitate, immediately marking the formation of water-stable α-FAPbI_3_. The phase stability of FAPbI_3_ is governed by kinetic and thermodynamic factors. A kinetically controlled product is stable at room temperature as it requires the lowest energy barrier. In contrast, a thermodynamically controlled product is the most stable at high temperatures and features the lowest surface energy. In light of this definition, δ-FAPbI_3_ synthesized at low temperature is a kinetically controlled product, while α-FAPbI_3_ trapped with amines possesses lower surface energy and falls into a thermodynamically controlled product. In a previous study, synthesizing a CsPbI_3_ incorporating dimethylammonium via controlling the interplay between kinetics and thermodynamic control explicitly demonstrated this phenomenon [41]. Likewise, α-CsPbI_3_ was stabilized by nanocrystals (NCs) much below the transition temperature for bulk materials [42]. Additionally, it is known that H^+^ and I^−^ are hard acids and soft bases, respectively, making a base-acid interaction between H^+^ of amines and I^−^ of [PbI_6_]^4−^ weaker. Such weak interactions are responsible for destabilizing perovskites in polar solvents, including water [43]. These studies propel us to find a suitable way to stabilize FAPbI_3_ in water using amines.

The crystal structure of the perovskites was confirmed by obtaining high-power powder X-ray diffraction (HP-PXRD), as shown in Figure 1a. It is observed that δ-FAPbI_3_ adopts a hexagonal phase where other perovskites have a α-phase. The diffraction peaks at 11.9° (010) and 16.3° (011), with other details in Figure 1a and Appendix A, confirm the phase of pure δ-FAPbI_3_ [44]. It is interesting to report that these δ-FAPbI3 characteristic peaks are generally absent in nearly all the perovskites with highly α-FAPbI_3_ features. These missing characteristic peaks also indicate that amines have impregnated inside the crystal structure, and now the FAPbI_3_ may not be pure but rather a hybrid of amine(s) cations at the A site of the perovskites, making the resulting cation larger in size (Appendix A) than the reported pure FAPbI_3_ NCs. Interestingly, as the nature of amines changes, the position of the diffraction peak changes. For instance, the peak position at ~14.0° and lower angle shifted peak to 11.62° (11.9° in δ-FAPbI) in (oleylamine) reflect α and δ-FAPbI_3_, respectively. Lower-angle shifts also mark a larger cation, which ascertains that oleylamine has been integrated into the FA^+^ cations. Quite an identical explanation could be given to all the remaining amines. The emergence of a new peak around ~6.0° could not be ascribed to any known perovskite or Pb-X phase, but in literature, it is reported presumably to be a complex of Pb halide and DMF/amine complexes [45,46]. The XRD pattern in (4-4′-methylenebis) seems slightly amorphous in nature. This could be explained by the fact that 4-4′-methylenebis (cyclohexylamine) belongs to a cycloaliphatic class of amines. This amine is considered a bulky amine (steric hindrance), and its incorporation in the FAPbI_3_ may not align well with the crystal structure, leading to excessive disorder in the basic structure, which is well reflected in its amorphous nature in XRD.

Thermogravimetric analysis (TGA) was carried out in the range of 40–900 °C under N_2_ gas flow (Figure 1b). The decomposition onset temperature of all the amines falls between ~200 and 250 °C, with the exception of oleylamine (>250 °C) and about δ-FAPbI_3_ (290 °C) [14]. The early-onset decomposition temperature for amines is not far to seek, as they are loosely bound to the perovskites and evaporate quickly. However, oleylamine, a long-chain unsaturated fatty amine, is hydrophobic. Due to its hydrophobicity, it resists easy decomposition in the presence of moisture and temperature. In general, all the amines show decomposition in a broad range of temperatures (260–450 °C) due to the loss of amines-iodide and FAI. However, all the perovskites show their second decomposition at ~440 °C, associated with the decomposition of PbI_2_. TGA data reveals that α-FAPbI_3_ remains stable up to 250 °C.

Morphological revelation under SEM shows contrasting features in different amines (Figure 2a–f). SEM along with the EDX image of δ-FAPbI_3_ (Figure 2a and Appendix A) shows sheet-like morphologies within a few micrometers in size with a low magnification image, as shown in Appendix A. Furthermore, the EDX mapping indicating the presence of N (Figure 2g) confirms that the addition of amines stabilizes the α-FAPbI_3_ structure. In the presence of moisture, 3D-structured perovskites are converted to 2D structures before decomposition to PbI_2_. The presence of amines plays a decisive role in this transformation. In the presence of moisture, these amines are converted to their salts, transforming the perovskites into 2D sheet-like structures [47]. These NSs morphologies assist in better encapsulation of the perovskite structure, restrict the rapid degradation of perovskites in the presence of moisture, and enhance their water stability for practical application.

To reveal the electronic structures of the perovskites, their X-ray photoelectron spectroscopy (XPS) data was collected. XPS data has been the most widely used surface analysis technique. XPS measurements can reveal the chemical composition and electronic states of elements in surface-mediated processes, including redox reactions, oxygen evolution reactions, dissolution/precipitation, encapsulations, and various other deposition-type reactions. XPS data convolutions have been obtained by Shirley background subtraction to assign various peaks (Figure 3). The XPS peak positions for Pb4f_7/2_ and Pb4f_5/2_ in δ-FAPbI_3_ and α-FAPbI_3_ show variations in their peak positions (Figure 3a,d and Appendix A). These variations in XPS spectra indicate that electronic structure changes after the incorporation of various amines. To avoid complexity and make the discussion clear, we have just considered oleylamine variations in XPS spectra with δ-FAPbI_3_, while other amines are shown in Appendix A. The binding energies of Pb4f_7/2_ and Pb4f_5/2_ are located at (138.18/137.48 e.V.) and (142.98/142.38) for δ/α-FAPbI_3_, respectively. While small peaks at 140.08 e.V. (C1) and 144.98 e.V. (C2) associated with Pb4f_7/2_ and Pb4f_5/2_, respectively, indicate the formation of carbonates of lead, Pb(CO_3_)_2_ [48], reflecting the adsorption of atmospheric CO_2_ [49]. It could be further seen that Pb exists in its varying oxidation states (Pb^4+^/^2+^), and thus, its octahedral structures may not be in perfect 3D geometry; nonetheless, the δ-phase becomes the most preferred avenue. It could also be seen that these small peaks (i.e., C1 and C2) remain absent in α-FAPbI_3_, indicating the incorporation of amines, which thereby prevents the attack of atmospheric CO_2_ or moisture (H_2_O). PbO or Pb(CO_3_)_2_ formation is suppressed due to the strong charge acceptance capability of I via donor-acceptor pair-forming complexes between I and Pb (Figure 3b,e). The Pb (6p) orbital donates its excess unpaired valence electrons to electronegative I^−^, and in the process, it oxidizes to Pb^2+^ while iodine reduces to 2I^−^, resulting in a more stable α-FAPbI_3_ (Figure 3e). The Pb peaks at 137.48 e.V. and 142.38 e.V. correspond to Pb^2+^ present in a 2D layer of the α-FAPbI_3_ perovskite. In δ-FAPbI_3_, the peak at 400.38 e.V. (C8) and 402.18 e.V. (C9) corresponds to N-C amines and N-O-like species, respectively. Moreover, two peaks at 401.18 e.V. and 399.68 e.V. correspond to FA and surface-bound amines, respectively (Figure 3c,f).

To get insights on the optical properties of these synthesized α-FAPbI_3_ materials, their photoluminescence (PL) data was obtained. For the sake of simplicity and to avoid misalignments, we present only four amines, and the others are detailed in Appendix A. In general, PL spectra dictate the light-emitting and absorbing nature of the perovskite materials. The PL emissions for α-FAPbI_3_ show dominant emission around 796 nm (for 450 nm excitation); this is in agreement with a previous report [50]. However, another emission peak at ~581 nm (Figure 4a) indicates the 2D structure of the synthesized perovskite in the presence of oleylamine. However, as the nature of amines changed (from aliphatic to aromatic amines), the emission peaks and the dual emission behavior changed significantly. In ethylenediamine, dual emission peaks are noticed at 590 and 789 nm, reflecting the perovskite’s 2D and 3D layers. Meanwhile, in N-methylaniline, these peaks are at 544 and 805 nm, and in 2-aminoethanol, peaks appear at 586 and 791 nm. What is exciting to see from these PL spectra is that the emission peaks are shifting toward 3D structures as amines get more complex (from primary amines to aliphatic and aromatic). Interestingly, in 4-4′-methylenebis (cyclohexylamine), which is indeed a class of diamines, there is a single dominant emission peak at 790 nm and a lean peak at 588 nm. These indicate that all these amines stabilize the perovskites mostly in 3D structure, reflecting the dominance of 3D peaks over 2D structure. When treated with bulky amines, FAPbI_3_ is stabilized in their α-perovskite phase; a quite similar result has been shown where a bulky cation like phenethylammonium iodide restricts the FAPbI_3_ into a non-perovskite phase with a PL emission centered at ~800 nm [51]. From the lean PL peak at 588 nm in 4-4′-methylenebis, it could be inferred that these are low dimension (LD) phases quite common when sterically/bulky cations interact with 3D perovskites [51], like the one in 4-4′-methylenebis (cyclohexylamine) with FAPbI_3_.

Having dual emissions in the PL spectra allows us to analyze their optical properties under confocal microscopy (Figure 5a,b). Confocal microscopes are a strong tool for distinguishing between one- and two-photon excitons. We investigated the dual emission behavior in the case of 2 by confocal microscopy to justify the PL spectra. The same particle emits dual emissions upon excitation with a 458 nm laser, with one being green and the other being pink (Figure 5b). We also confirm our micrometer-sized synthesized particles from the confocal microscopy analysis.

We further explore the structural stabilizing effect of amines through obtaining XRD patterns after 3 months. To our surprise, we found that the XRD intensity of oleylamine declined slightly even after 3 months. It is essential to state that, though amine stabilizing is a strong tool, complete eradication of degradation is a daunting task ahead. Over a prolonged period, particularly in an environment containing degrading factors (moisture, heat, and light), perovskite material may still experience degradation to a certain level. The amines present in the perovskite crystals control the grain size, and the reduced grain size could be seen as the amines get longer or more complex (Figure 2). These long-chain amines suppress perovskites’ omnidirectional growth and restrict them to reduced grain size [52]. Furthermore, the incorporation of amines also arrests the phase transition from α to δ-FAPbI_3_ and stabilizes α-FAPbI_3_.

## 4. Conclusions

In summary, we show for the first time that the aqueous stability of α-FAPbI_3_ could be achieved by tuning the crystal structure with various amines following a top-down approach. The structural study using XRD reveals that amine-mediated crystal structures show variations among themselves on the basis of the varying nature of amines. The long-term stability of α-FAPbI_3_ perovskite is ascribed to different amines (such as aliphatic long chains, bulky cations, aromatics, and so on), which restrict the penetration of the decomposing agent into the crystal structure, as can be seen from the XRD patterns. SEM morphologies are utilized to observe changes in grain size, surface features, and overall morphology of α-phase FAPbI_3_ NCs bonded to various amines. These stabilization studies are necessary to find and explore several moieties that can potentially restrict perovskite from going to its non-perovskite phase, opening a wider application field for their applications. In the continuation of this research, we are investigating different amines as potential stabilizers for α-FAPbI_3_ perovskite. We aim to broaden the scope of stabilizing agents and contribute a simple yet effective strategy to enhance the stability of the α-FAPbI_3_ phase. We anticipate that our approach could complement existing strategies within the perovskite field, ultimately paving the way for designing innovative materials for various energy applications.

## Data Availability

All the experimental data presented within this article, along with the Appendix A, will be made available from the authors upon reasonable request.

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
