# Peer review of "Achieving Order in Disorder: Stabilizing Red Light-Emitting α-Phase Formamidinium Lead Iodide"

_nanomaterials, 2023, doi:10.3390/nano13233049_

Round 1

Reviewer 1 Report

Comments and Suggestions for Authors

Manuscript entitled “Achieving order in disorder: Stabilizing red light-emitting α-2 phase formamidinium lead iodide” by Aditya Narayan Singh, Atanu Jana, Manickam Selvaraj, Mohammed A. Assiri and Kyung-Wan Nam presents synthesis and characterisation  of formamidinium lead iodide (FAPbI3) halide perovskite. Authors attempt to stabilize the perovskite alpha phase, which is attractive from point of photovoltaics.

The manuscript written in manner rather difficult to read. English language should be checked and corrected. For example: “Few things are agreed so well in the research community as the fact that the global atmosphere will continue to grow if anthropogenic emissions particularly CO2-are not strictly reduced to zero”. What does it mean “the global atmosphere will continue to grow” ? Another example “This awareness has led to several international pledges, the prominent one is the Paris Agreement to keep global temperature well below 2 °C this century”.

The manuscript could be suitable for publication in the journal after revisions according comments presented below.

Rows 58-59: During its synthesis, FAPbI3 exhibits different morphologies with variations in temperature, namely, at room temperature (< 300 K), a trigonal non-perovskite δ-phase. It would more precise to write :… exhibits different phases …”?

Rows 63-64: “Because of this narrow band gap and thermal stability α-phase FAPbI3 is considered a potential …” and in the next sentence it written “…this phase is metastable and thus stabilization at room temperature is necessary …”.

What was the difference between synthesis protocols for delta (2.2) and alpha (2.3) phases?

Rows 152-153: The samples for XRD analysis were prepared by conventional technique by applying a thick layer of cathode powder onto a glass substrate. What does it mean “cathode powder” ?

Row 160: “Particle morphologies were revealed under SU8220 …” Does it mean that authors putted the particles under the microscope?

Rows 177-182: There is no need to repeat the synthesis protocol at the beginning of the Results and discussion section.

Figure 1a is very uninformative. A reader should only believe that XRD pattern no 1 presents delta phase and the rest patterns alpha phase. The peaks on the patterns must be indexed so that thr reader could check which phases were present, did powders present pure phase. The 2 theta range of 5-35 degrees could be sufficient.

Figure 2. (g) SEM-EDS mapping of α-FAPbI3. Figure 2g does not present SEM-EDX mapping but just x-ray spectrum of the sample.

The Figure 2g contains table, in which one can see that sample amounts 41.46 at.% of iodine and only 7.17 at.% of lead. How it could be explained, if according chemical formula the quantity of iodine must be higher than that of lead only by factor of 3 (in this case should be 21.51 at.% instead of 41.46 at.% presented in the table)?

Rows 253-254: “XPS data has been a widely used tool to reveal exciting chemistry on surface mediated processes …” What does it mean “exciting chemistry” ?

Figure 3. XPS spectra of what samples are presented in Figure 3a-c and Figure 3d-f ? The caption of Figure 3 tells that spectra a-c is for delta FAPbI3, so does it mean that those spectra is for the sample 1 ? If yes, then according XRD pattern it should be pure delta phase, so why XPS spectra in Figure 3 a and b show peaks of alpha and delta phases?

Figure 5: (c) Structural stability after 3 months. It would be better to write: (c) XRD patterns of sample 2 evidencing structural stability.

Rows 338-339: “…restricts the attacking degradation agent to penetrate the crystal structure as evident by long term XRD spectra.” It would be better to write: “... restricts the penetration of the decomposing agent into the crystal structure, as can be seen from the XRD patterns.“ Spectrum shows radiation intensity as a function of radiation wavelenght or frequency while XRD pattern presents radiation intensity as a function of diffraction angle and the radiation wavelength is constant, so it is not spectrum.

Rows 339-341: “Incorporation of amines in the crystal structures could be well supported virtually by SEM morphologies which indicate a change in their grain size.”

SEM images are not evidence of the incorporation of amines into the crystalline structure. The size of the grains or crystals is not directly related to the crystalline structure.

Rows 343-344: “In addition, we also We are in the process of understanding this from the machine learning perspective and using various other amines and confirm our proposition.”

This sentence must be revised.

Comments on the Quality of English Language

The manuscript written in manner rather difficult to read. English language should be checked and corrected.

Reviewer 2 Report

Comments and Suggestions for Authors

This manuscript reported the synthesis of water-stable red light-emitting α-phase FAPbI3 nanocrystals (NCs) using 5 different amines to overcome these intrinsic phase instabilities. The structural, morphological, and electronic characterization were obtained using XRD, FESEM and XPS. It can be accepted after minor revisions.

1. In Figure 1a, the crystal planes of the diffraction peaks should be marked.

2. In Figure 1a and 1b, the name of the samples should be used, rather than 1, 2, 3,4,5,6.

3. In the figure caption of Figure 2 and Figure 4, the name of the samples should be used, rather than 1, 2, 3,4,5,6.

4. In Figure 4, PL should be used, rather than Pl.

5. In the Conclusions, the sentence of “In addition, we also We are in the process of understanding this from the machine learning perspective and using various other amines and confirm our proposition.” is wrong. Please revise it.

6. Some recent reports regarding the application of perovskite should be cited, such as J. Mater. Chem. C, 2023, 11, 25402551; J. Phys. Chem. Lett. 2023, 14, 52495259; Sol. RRL 2023, 7, 2300038.

Comments on the Quality of English Language

Minor editing of English language required.

Reviewer 3 Report

Comments and Suggestions for Authors

Aditya Narayan Singh et al. reported the stabilizing red light-emitting α-phase formamidinium lead iodide. Although this work is very important, here are some questions to be addressed as follows,

(i)The font size of some parts of the paper is different from the whole (e.g. Table 1), please check carefully.

(ii)For compound 5, the PL peak described in the paper (at 761 nm) seems to be inconsistent with the peak in the figure.

(iii)The article mentions the use of PL spectroscopy to understand the absorption and emission properties of chalcogenides, but in fact mainly shows the emission properties and does not discuss the absorption properties of materials 3, 4, 5, and 6, so it is suggested that PLE spectroscopy or UV-visible spectroscopy be supplemented.

(iv)Please standardize abbreviations, for example, the “l” in figure 4 “Pl intensity” should be capitalized.

(V)The latest references about phase formamidinium lead iodide need to be added and updated.

Based on the above analysis, the manuscript is published after minor revision.

Comments on the Quality of English Language

Aditya Narayan Singh et al. reported the stabilizing red light-emitting α-phase formamidinium lead iodide. Although this work is very important, here are some questions to be addressed as follows,

(i)The font size of some parts of the paper is different from the whole (e.g. Table 1), please check carefully.

(ii)For compound 5, the PL peak described in the paper (at 761 nm) seems to be inconsistent with the peak in the figure.

(iii)The article mentions the use of PL spectroscopy to understand the absorption and emission properties of chalcogenides, but in fact mainly shows the emission properties and does not discuss the absorption properties of materials 3, 4, 5, and 6, so it is suggested that PLE spectroscopy or UV-visible spectroscopy be supplemented.

(iv)Please standardize abbreviations, for example, the “l” in figure 4 “Pl intensity” should be capitalized.

(V)The latest references about phase formamidinium lead iodide need to be added and updated.

Based on the above analysis, the manuscript is published after minor revision.
